# Theoretical Design and Experimental Validation of a Nonlinear Controller for Energy Storage System Used in HEV

**Zakariae El Idrissi \***, **Hassan El Fadil, Fatima Zahra Belhaj, Abdellah Lassioui,**
**Mostapha Oulcaid and Khawla Gaouzi**

LGS Laboratory, ENSA, Ibn Tofail University, Kenitra BP 242, Morocco; elfadilhassan@yahoo.fr (H.E.F.);
fz.blhj@gmail.com (F.Z.B.); abdellah.lassioui@uit.ac.ma (A.L.); oulcaid02@gmail.com (M.O.);
khawla.gaouzi@gmail.com (K.G.)

**\*** Correspondence: zakariae.elidrissi@gmail.com; Tel.: +21-26-4035-6189

**Abstract:** This work presented a nonlinear control for a reversible power buck–boost converter (BBC) in order to control energy storage in a supercapacitor (SC) used in hybrid electric vehicles (HEV). The aim was to control a power converter in order to satisfy the following two requirements: (i) perfect tracking of SC current to its reference signal and (ii) asymptotic stability of the closed-loop system. The two objectives were achieved using an integral sliding mode control. In order to validate the proposed approach, an experimental prototype was built. The controller was integrated into dSPACE prototyping systems using the DS1202 card. It was clearly shown, using formal analysis, simulation, and experimental results, that the designed controller metall the objectives, namely, the stability of the system and the control of the current at its reference.

**Keywords:** hybrid electric vehicle (HEV); supercapacitor (SC); integral sliding mode control (ISMC); Lyapunov theory; dSPACE DS1202 real-time control (RTC) card; experimental validation

---

## 1. Introduction

Nowadays, much research has been undertaken on technologies for future vehicles. Among these technologies, the hybrid electric vehicle (HEV) is an efficient and promising solution [1–3]. The hydrogen-based HEV is a concept that combines between two sustainable and clean fields because the main source of energy for this vehicle is based on hydrogen, which is licensed as renewable resources [4,5], and the goal is to contribute to reduce $CO_2$ emissions and global warming effects, to present a mean of transportation capable to concur the classical vehicles in performances with zero emissions [6]. Owing to their economic and environmental benefits, the State invites and encourages by scholarships and by tax exonerators the researchers and industrialists to combine them and to dig in the field of the hybrid automobile in order to develop the infrastructure of the HEV [7].

Hybrid electric vehicles have two sources of energy, namely, the main energy source (MES) and auxiliary energy source (AES). The MES based on a fuel cell provides autonomy for the normal operation of the vehicle, and the AES is used to supply electrical energy through a buck–boost converter (BBC) to the direct current (DC) bus at the time of acceleration of the vehicle or to recover energy when applying brakes. As a result, fuel cell (FC) vehicles have the potential to significantly improve fuel economy and can be more efficient than traditional internal combustion engines [8–10].

The FC-based energy source is not always sufficient to meet the requirements of an electric vehicle [11]. In order to provide the necessary power during transient phases, such as starting, acceleration, or sudden changes in vehicle speed, a supercapacitor (SC) bench is required for the HEV [12–14]. This SC bench also allows us to recover energy when the brakes are applied in a vehicle.

To ensure the exchange of energy between the different elements of the vehicle system, a robust control law is needed, in order to control the different converters of the studied system. There are several topologies for HEV [15,16]. The topology of the system studied is shown in Figure 1, which is generally called a hybrid energy storage system (HESS).

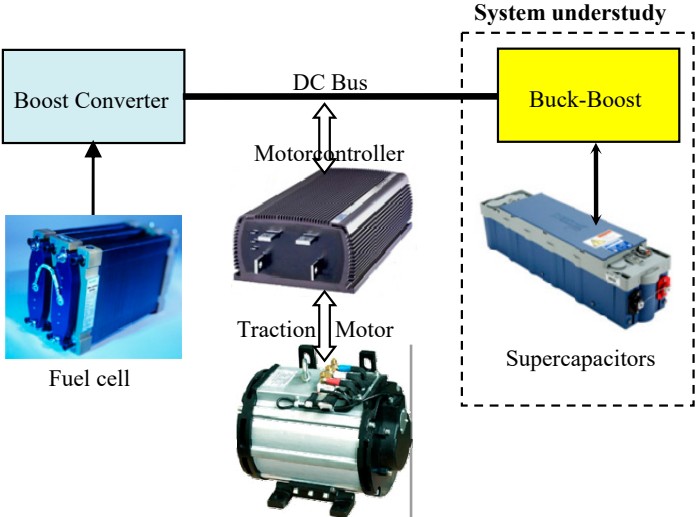

**Figure 1.** The power circuit of a typical hybrid electric vehicle.

The full cell is connected with the DC bus through the unidirectional boost power converter. The SC module is connected with the DC bus via the reversible buck–boost current converter to ensure the exchange of energy between the organs of the system (FC, SC, and motor). The motor is connected to the DC bus through the three-phase reversible current inverter to recover electrical energy when the brakes are applied in HEV.

Lately, many control strategies for power converters have been proposed. In the research of Ouyang, M. and Yang, W. [17,18], by the sliding mode control, a controller based on the Lyapunov function has been proposed to regulate the SC and battery currents. The references mentioned above relate only to the permanent regime of the system, and the authors have neglected the study of the transient regime, that is to say, the overshoot and the response time of the system. For the study of the stability of the system, they are based on the convergence of the error in steady-state. However, many nonlinear control strategies take into consideration the improvement in the performance of the transient regime and its balance [19–21]. In the research of Song, H. [19], the authors have proposed a robust dynamic surface controller to improve the performance of nonlinear systems. In the research of Liu, Q. [20], the authors have proposed a controller that estimates the unknown parameters of the electric vehicle system; this controller is based on an adaptive law. However, the results of [20] have a significant tracking error from the SC current $i_{sc}$ to the reference current $I_{scref}$ and also show significant ripple in the current $i_{sc}$, which influences the DC–DC bus voltage. Indeed, the robustness of the control proposed by [20] is low. The authors of [21] have presented an interleaved two-phase bidirectional DC–DC converter topology to control the SC current; this topology includes a small number of components based on a classical Proportional-Integral-Derivative (PID) control through the linearization of the nonlinear system. But the results of [21] show a significant ripple in the SC current, which introduces measurement and control errors. This is due to the fact that the system parameters are incorrectly dimensioned. Indeed, the control law proposed by [21] is not robust.

This paper dealt with the modeling of a reversible power buck–boost converter and then a nonlinear control strategy, in order to control the current of the SC in both charge and discharge cases. Finally, the experimental results by dSPACE DS1202 card from this study were presented. The contributions of the proposed control system were the simplicity of the process for controlling energy storage in an HEV. Then, the perfect control of charge and discharge current and the system

dynamics were improved compared to [20,21] due to the nonlinear control by integral sliding mode control (ISMC). On the other hand, the stability of the system and the robustness of the control law were proven by experimental validation. Finally, the methodology proposed in this paper could be used in an HEV.

The flow of this document is organized as follows: Section 2 is devoted to the modeling of a reversible power buck–boost converter; the design of the controller and the closed-loop analysis are presented in Section 3; the controller performance is illustrated by numerical simulation and by experimental validation in Section 4; Section 5 provides a conclusion to the document.

## 2. Storage System Presentation and Modeling

### 2.1. Storage System Presentation

Figure 2 shows the studied part of the HEV. It consisted of a 24 V DC bus provided by an FC; it was the main source of the vehicle, which was connected to the DC bus via a DC–DC converter reversible in current. The SC was used as an auxiliary source: it supplied transient power demand and peak loads required during acceleration and deceleration of HEV.

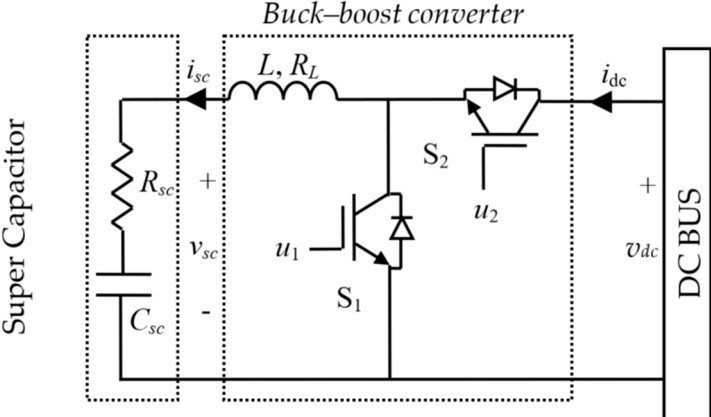

**Figure 2.** Supercapacitor energy storage system.

$L$ represents the inductor used for energy transfer and filtering. The inductor size is classically defined by switching frequency and current ripple [22]. The converter is driven by means of binary input signals $u_1$ and $u_2$ applied on the gates of the two Insulated Gate Bipolar Transistor (IGBTs) $S_1$ and $S_2$, respectively. The resistance $R_L$ represents the equivalent series resistance (ESR) of the inductor. The SC is represented by its capacity $C_{sc}$ and by its series resistance $R_{sc.}$

### 2.2. Modeling of a Reversible Power Buck–Boost Converter

The buck–boost converter could operate as a boost converter or a buck converter. Indeed, in the discharging mode of the SC ($i_{sc} < 0$), the converter operated as a boost converter, and in the charging mode of SC ($i_{sc} > 0$), it operated as a buck converter. As our goal was to enforce the SC current $i_{sc}$ to track its reference $I_{scref}$ provided by the energy management system, in order to control this converter, we have defined a binary variable k as follows:

$$k = \begin{cases} 1 & if \quad I_{scref} < 0 \quad \text{(Boost mode)} \\ 0 & if \quad I_{scref} > 0 \quad \text{(Buck mode)} \end{cases} \tag{1}$$

- Boost mode operation ($k = 1$)

In this case, the control input signal $u_2$ was fixed to zero ($u_2 = 0$), and $u_1$ was a PWM variable input. From inspection of the circuit, shown in Figure 2, and taking into account that $u_1$ could take the binary values 1 or 0, the following bilinear switching model could be obtained:

$$\frac{di_{sc}}{dt} = -(1 - u_1)\frac{v_{dc}}{L} - \frac{R_L}{L}i_{sc} + \frac{v_{sc}}{L} \tag{2a}$$

$$i_{dc} = (1 - u_1)i_{sc} \tag{2b}$$

- Buck mode operation ($k = 0$)

The control input signal $u_1$ was fixed to zero ($u_1 = 0$), and $u_2$ acted as the Pulse Width Modulation (PWM) variable input. From Figure 2, and taking in account that $u_2 \in [0,1]$, the Buck model could be obtained by:

$$\frac{di_{sc}}{dt} = -u_2\frac{v_{dc}}{L} - \frac{R_L}{L}i_{sc} + \frac{v_{sc}}{L} \tag{3a}$$

$$i_{dc} = u_2 i_{sc} \tag{3b}$$

The next step was to get a global model for buck–boost converter. From Equations (2a) and (2b) and Equations (3a) and (3b), we could obtain the following global model of BBC by:

$$\frac{di_{sc}}{dt} = -[k(1 - u_1) + (1 - k)u_2]\frac{v_{dc}}{L} - \frac{R_L}{L}i_{sc} + \frac{v_{sc}}{L} \tag{4a}$$

$$i_{dc} = [k(1 - u_1) + (1 - k)u_2]\,i_{sc} \tag{4b}$$

Equation (4a) could be rewritten as follows:

$$\frac{di_{sc}}{dt} = -u_{12}\frac{v_{dc}}{L} - \frac{R_L}{L}i_{sc} + \frac{v_{sc}}{L} \tag{5a}$$

where $u_{12}$ is the control input of BBC defined as follows:

$$u_{12} = k(1 - u_1) + (1 - k)u_2 \tag{5b}$$

In order to establish the control law of this BBC, by averaging the model (5a) over a switching period, the average model was:

$$\frac{dx_1}{dt} = -\mu_{12}\frac{v_{dc}}{L} - \frac{R_L}{L}x_1 + \frac{v_{sc}}{L} \tag{6}$$

where $x_1$ the average value of the SC current ($x_1 = <i_{sc}>$), and $\mu_{12}$ is the duty cycle, i.e., average values of the binary control input $u_{12}$ ($\mu_{12} = <u_{12}>$), which takes values in [0,1]. The generation of effective control input signals $u_1$ and $u_2$ from $u_{12}$ is represented in Figure 3.

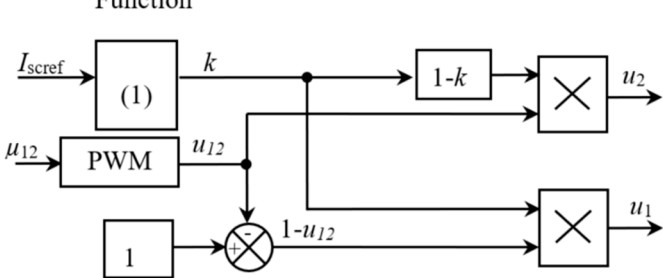

**Figure 3.** Block diagram of input signals $u_1$ and $u_2$ generation.

## 3. Storage Sliding Mode Control and Stability Analysis

Our contribution consisted of implementing a robust nonlinear controller to control the charge and discharge current of the SC, for an objective of protecting the SC against misuse, allowing us to make good use of the normal functioning of the SC and to extend its life as much as possible [23].

### 3.1. Control Objective

In order to define the control strategy, the first one had to establish the control objectives, which could be formulated as follows:

(i).  Monitoring of the supercapacitor current up to its reference,
(ii).  Asymptotic stability of the system.

### 3.2. Sliding Mode Control

The sliding mode technique changed the structure of the controller in response to the changing state of the system. This was realized by the use of a high speed switching control, forcing the trajectory of the system to move to and stay in a predetermined surface, which is called a sliding surface. In sliding mode, a system's response remained insensitive to parameter variations and disturbances. The sliding mode control technique could be a possible option to control this kind of circuits. The following trajectory was defined by:

$$S = K_1 e + K_2 \int_0^t e \, dt \tag{7}$$

where $K_1$ and $K_2$ are sliding surface coefficients, and $I_{scref}$ is the current reference, and $e = x_1 - I_{scref}$ is the surface error.

The derivative of the surface (Equation (7)) was given by:

$$\dot{S} = -\mu_{12} K_1 \frac{v_{dc}}{L} + \left( K_2 - K_1 \frac{R_L}{L} \right) x_1 + K_1 \frac{v_{sc}}{L} - K_2 I_{scref} \tag{8}$$

In this paper, we considered that the control law $\mu_{12}$ consisted of two components: an equivalent component $\mu_{12eq}$ and a nonlinear component $\mu_{12n}$:

$$\mu_{12} = \mu_{12eq} + \mu_{12n} \tag{9}$$

The equivalent control component constituted a control input, which, when exciting the system, produced the motion of the system on the sliding surface whenever the system was on the surface. The existence of the sliding mode implied that $\dot{S} = 0$ [24,25].

It followed that the equivalent control $\mu_{12eq}$ could be obtained, using Equation (8) and the fact that in sliding mode, $\dot{S} = 0$ as follows:

$$\mu_{12eq} = \frac{L}{v_{dc}} \left( \left( \frac{K_2}{K_1} - \frac{R_L}{L} \right) x_1 + \frac{v_{sc}}{L} - \frac{K_2}{K_1} I_{scref} \right) \tag{10}$$

The objective of the second component $\mu_{12n}$ was to ensure the equilibrium $S = 0$ to be globally asymptotically stable. To this end, we considered the following positive definite Lyapunov function:

$$V = \frac{1}{2} S^2 \tag{11}$$

The derivative of the Equation (11) was given by:

$$\dot{V} = \dot{S} S \tag{12}$$

This equation gave, using Equations (8) and (9):

$$\dot{V} = S\left(\begin{array}{c} -\mu_{12n}K_1\frac{v_{dc}}{L} - \mu_{12eq}K_1\frac{v_{dc}}{L} + \\ \left(K_2 - K_1\frac{R_L}{L}\right)x_1 + K_1\frac{v_{sc}}{L} - K_2 I_{scref} \end{array}\right) \tag{13}$$

which, in turn, gave, using Equation (10):

$$\dot{V} = S\left(-\mu_{12n}K_1\frac{v_{dc}}{L}\right) \tag{14}$$

This equation clearly showed that the nonlinear component $\mu_{12n}$ could be chosen as follows:

$$\mu_{12n} = \frac{L}{K_1 v_{dc}}\lambda S \tag{15}$$

where $\lambda$ is a positive design parameter.

Indeed, with this choice, Equation (14) became:

$$\dot{V} = -\lambda S^2 \tag{16}$$

which is negative definite.

This ensured that the equilibrium $S = 0$ was globally asymptotically stable (GAS).

Finally, combining Equations (9), (10), and (15), the sliding mode control law of the system was obtained:

$$\mu_{12} = \frac{L}{v_{dc}}\left[\frac{K_2}{K_1}e + \frac{1}{K_1}\lambda S - \frac{R_L}{L}x_1 + \frac{v_{sc}}{L}\right] \tag{17}$$

### 3.3. The Limitations of SMC Technique

Like all nonlinear controllers in the literature [26,27], there were advantages and disadvantages. The key point of the sliding mode control law was robustness, but it was also characterized by a problem of chattering on the sliding surface in the case where the reference was very frequent.

Practically, the implementation of such discontinuous controllers was characterized by the phenomenon of chattering. Chattering could be reduced by dividing the control into continuous and switching components so as to reduce the amplitude of the switching one [28].

## 4. Simulation and Experimental Results

The performances of the proposed nonlinear controller were illustrated by simulation and experimental results.

### 4.1. System Characteristics

The controlled system characteristics are listed in Table 1.

**Table 1.** Parameters of the Controlled System.

| Parameter | Value |
|---|---|
| Inductance $L$ | 4 mH |
| Inductances ESR, $R_L$ | 620 mΩ |
| Supercapacitor, $C_{sc}$ | 500 F |
| Supercapacitor ESR, $R_{sc}$ | 2.1 mΩ |
| Switching frequency, $f_{simulation}$ | 25 kHz |
| Switching frequency, $f_{experimental}$ | 15 kHz |

The type of transistor used in the simulation was a MOSFET transistor. According to the Datasheet of this transistor, the current/frequency $I_D(f_s)$ graph allowed us to choose the suitable switching

frequency for our system, which was 25 kHz. Likewise, for the experiment, the type of transistor used was an IGBT transistor under the reference IG15 and its control box under the reference IG10. According to the Datasheet for this transistor, its suitable switching frequency was 15 kHz.

*4.2. Simulation and Experimental Bench for SCSS Control*

The technology of dSPACE via MicroLabBox DS 1202 has simplified the implementation of the control law by the link between control Desk® and Matlab®/Simulink®, to easily test the systems or measure its quantities (Voltage, Current). The key point of this technology is the real-time control (RTC) process of the system. dSPACE systems are the solution for the development of embedded software in the automotive, aerospace, and industrial control [29,30].

The simulation bench of the SC energy storage system control is described in Figure 4 and was simulated using the MATLAB®/software®. In this figure, $\mu_{12}$ is the control law (the duty ratio), $u_1$ and $u_2$ are the binary input signals, $i_{sc}$, $V_{sc}$, and $V_{dc}$ are the measured variables, and $I_{\mathrm{scref}}$ is the SC current reference.

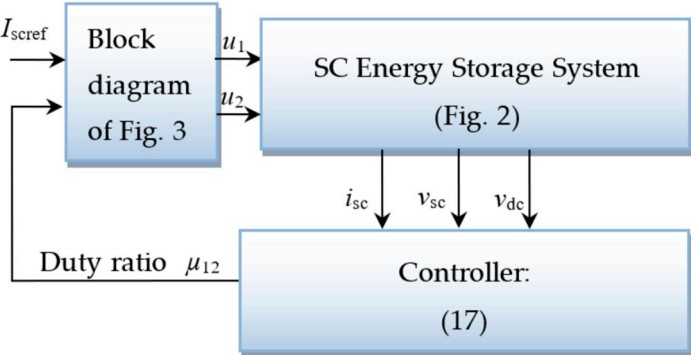

**Figure 4.** Simulation bench for supercapacitor storage system (SCSS) control.

The experimental test bench of the supercapacitor storage system (SCSS) control is described in Figure 5 and was implemented using dSPACE 1202 and Control Desk®/software®. This validation was carried out in the LGS Laboratory, ENSA, Ibn Tofail University.

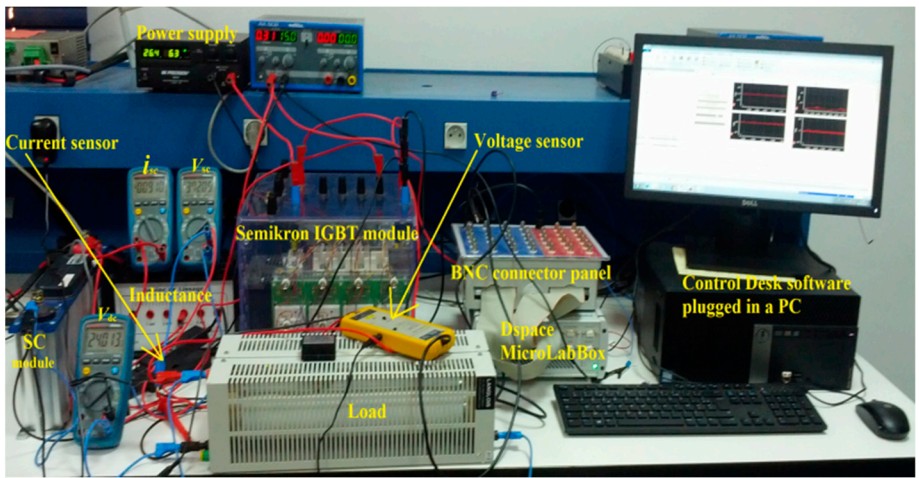

**Figure 5.** View of the experimental test bench for SCSS control.

To implement the proposed control system, it consisted essentially of:

- a power supply from BK Precision,
- a dSPACE DS1202 with Control Desk®/software® plugged in a Pentium 4 personal computer,

-     a Semikron IGBT module (SEMITEACH),
-     a 16 V supercapacitor module of Maxwell,
-     one ferrite inductance,
-     one Hall effect current sensor,
-     one voltage sensor,
-     a load.

The DC bus voltage was set to 24 V, and the initial voltage of SC was set to 2 V in the case of charging and 9 V in the case of discharging. The DC bus was represented by a voltage source in series with a full return diode. The load for this validation was a variable resistance characterized by 11.2 Ohm and 10 A.

The design control parameters were chosen as follows, which proved to be convenient: $\lambda = 1580$ and $K_1 = 7$ and $K_2 = 18$. Note that the parameters $K_1$ and $K_2$ and $\lambda$ were nonlinear control parameters of the reversible buck–boost current power converter. The values of its parameters were determined from the simulation.

### 4.3. Figures and Simulation Results

The simulation was performed under Matlab®/Simulink® over a reduced duration compared to the duration of the experiment because we were limited by the memory of the PC opposite to the number of points that were taken.

The resulting control performances of buck–boost power converter are shown in Figures 6–13.

Figures 6 and 10 illustrate the current measurement of $i_{sc}$ and its reference signal $I_{scref}$ for two scenarios: charging mode and discharging mode, respectively. In these figures, one could see that the controller behavior was satisfactory. Indeed, the SC current $i_{sc}$ perfectly tracked its reference $I_{scref}$. The overshoot was zero, the system response time was less than 0.7 s, and the signal ripple was tolerable, less than 0.08 A.

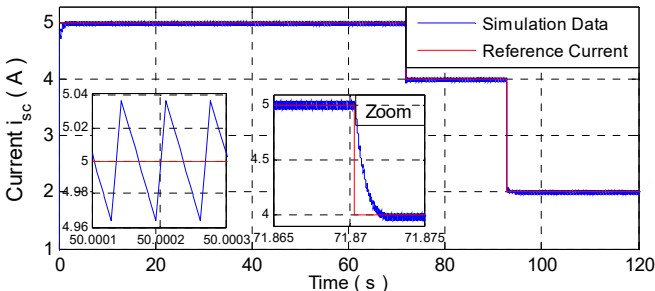

**Figure 6.** SC (supercapacitor) current $i_{sc}$ and its reference $I_{scref}$ with zoom.

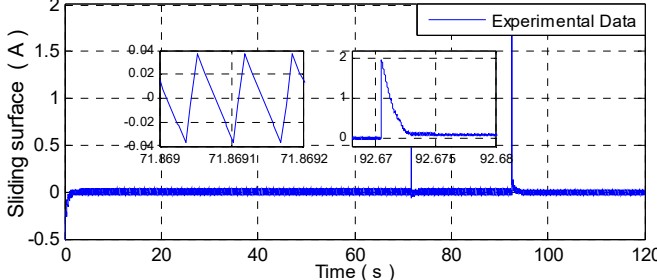

**Figure 7.** The sliding surface S with zoom.

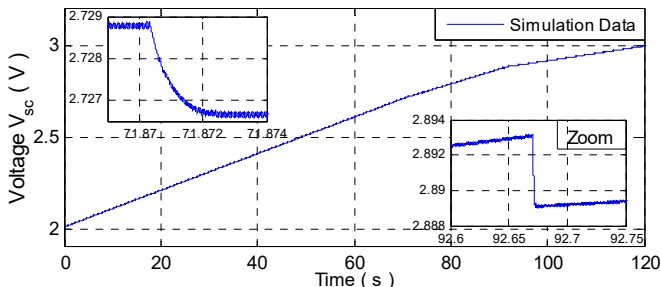

**Figure 8.** SC voltage $V_{sc}$ with zoom.

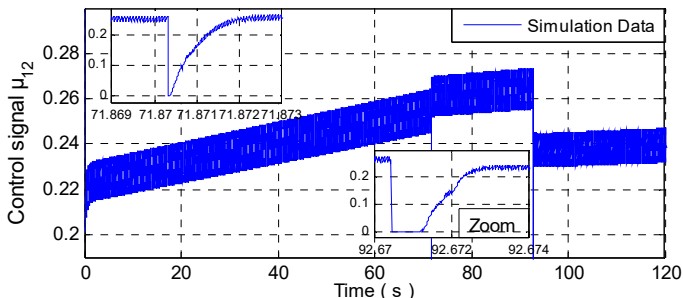

**Figure 9.** The control signal $\mu_{12}$ with zoom.

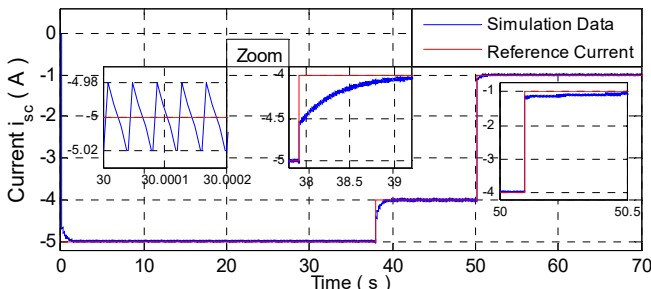

**Figure 10.** SC current $i_{sc}$ and its reference $I_{scref}$ with zoom.

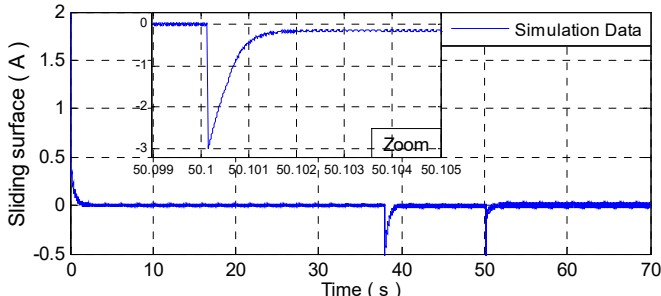

**Figure 11.** The sliding surface $S$ with zoom.

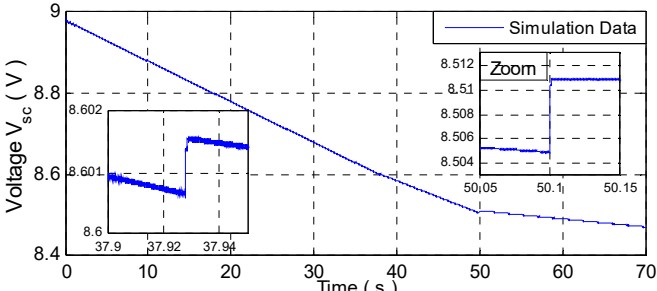

**Figure 12.** SC voltage $V_{sc}$ with zoom.

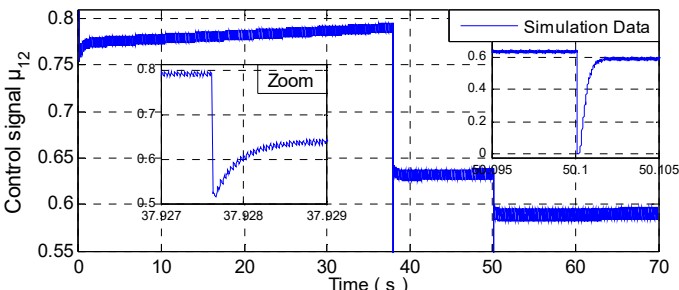

**Figure 13.** The control signal $\mu_{12}$ with zoom.

Figures 7 and 11 illustrate the trajectory of the sliding surface *S*. This figure clearly shows that the equilibrium *S* = 0 was attractive. The signal ripple was tolerable, less than 0.08 A. On the other hand, the enlarged parts in the figures show that the sliding surface was maintained at zero, despite variations in the value of the reference current. Hence, the robustness of our proposed control law ISMC.

Figures 8 and 12 show, respectively, charge and discharge mode of the SC voltage $V_{sc}$, and this mode was based on the value of $I_{scref}$. The value of the difference that existed in the $V_{sc}$ curve was a function of the deviation of $I_{scref}$ at the time of its change.

Figures 9 and 13 illustrate the control signals $\mu_{12}$. This signal was mainly a function of the values of $V_{sc}$ and $i_{sc}$ because the sliding surface was almost zero. On the other hand, the enlarged parts in the figures show the effect of the change of the value of the current on the control law because control law in Equation (17) was in function of the current $i_{sc}$ and $V_{sc}$.

■   Charging mode of SC (Buck operation, *k* = 0):
■   Discharging mode of SC (Boost operation, *k* = 1)

### 4.4. Figures and Experimental Results

The control system had been implemented in the dSPACE DS1202 via MicroLabBox and used with a real-time interface (RTI). The DS1202 was fully programmable from the Simulink® block diagram environment, and all input/output were configured graphically.

The experiment results of buck–boost power converter are illustrated in Figures 14–21.

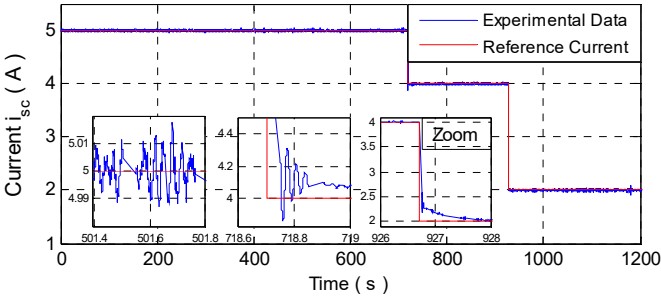

**Figure 14.** SC current $i_{sc}$ and its reference $I_{scref}$ with zoom.

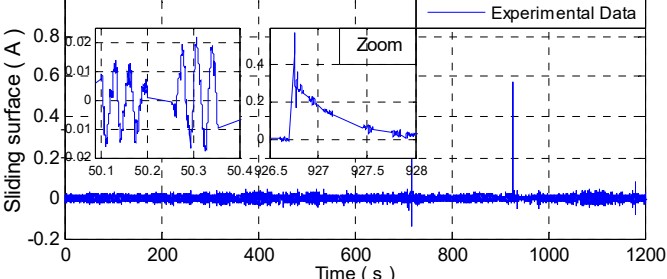

**Figure 15.** The sliding surface *S* with zoom.

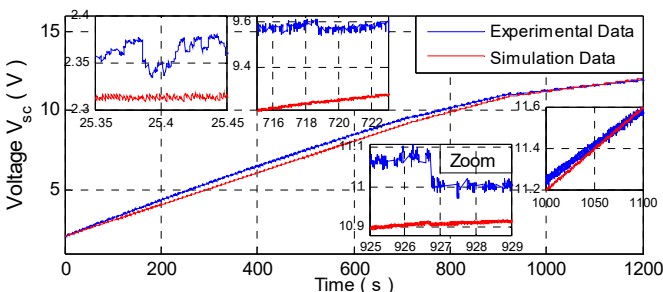

**Figure 16.** SC voltage $V_{sc}$ with zoom.

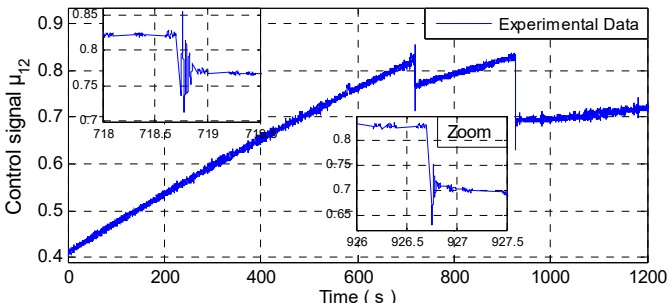

**Figure 17.** The control signal $\mu_{12}$ with zoom.

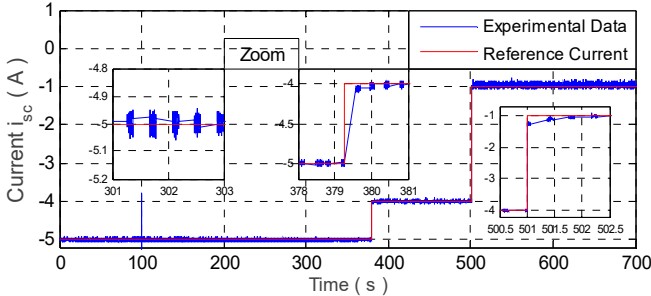

**Figure 18.** SC current $i_{sc}$ and its reference $I_{scref}$ with zoom.

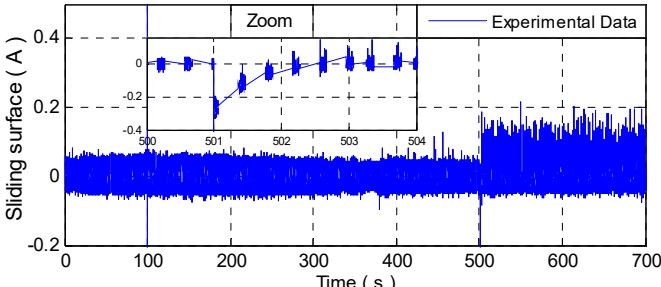

**Figure 19.** The sliding surface $S$ with zoom.

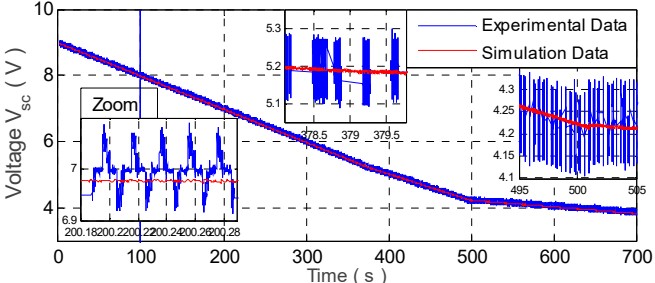

**Figure 20.** SC voltage $V_{sc}$ with zoom.

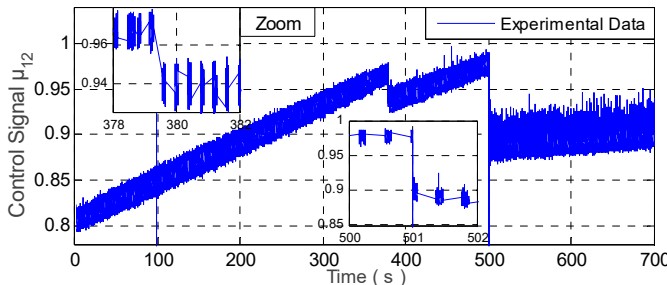

**Figure 21.** The control signal $\mu_{12}$ with zoom.

Figures 14 and 18 show that the controller behavior was satisfactory. Indeed, the SC current $i_{sc}$ perfectly tracked its reference $I_{scref}$. The overshoot was almost zero, the system response time was around 0.7 s, and the signal ripple was tolerable, less than 0.08 A due to measurement noise. Its results were better compared to the results of [21]. Indeed, the authors of [21] control the current indirectly by controlling the voltage, which generates significant undulations at the level of the current ($i_L = i_{sc}$) and, consequently, the aging of the SC. On the other hand, it was necessary to control the current of SC instead of the voltage to protect the SC.

Figures 15 and 19 illustrate the trajectory of the sliding surface *S*. This figure clearly shows that the equilibrium $S = 0$ was attractive. The signal ripple was tolerable, less than 0.04 A. On the other hand, the enlarged parts in the figures show that the sliding surface was maintained at zero, despite variations in the value of the reference current. Hence, the robustness of our proposed control law ISMC.

Figures 16 and 20 show, respectively, charge and discharge mode of the SC voltage $V_{sc}$, the difference between the experimental and simulation signals due to the voltage drop in the connection cables, and also the difference in the type of MOSFET transistor compared to IGBT. On the other hand, the enlarged parts in the figures show the effect of the internal resistance of the SC, in both cases, on the charge and discharge of the latter.

Figures 17 and 21 illustrate the control signals $\mu_{12}$. This signal was mainly a function of the values of $V_{sc}$ and $i_{sc}$ because the sliding surface was almost zero. On the other hand, the enlarged parts in the figures show the effect of the change of the value of the current on the control law because control law in Equation (17) was in function of the current $i_{sc}$ and $V_{sc}$.

We could note that:

-       The simulation and experimental results responded perfectly to the theoretical approach (ISMC, integral sliding mode control) used in this paper.
-       The results of the simulation of the reversible buck–boost current converter on Matlab®/Simulink® were identical to the experimental results that were taken by the dSPACE DS1202 card.

This would encourage researchers who do not have the equipment to conduct experimental validation by using our buck–boost model as a reference.

■       Charging mode of SC (Buck operation, $k = 0$)
■       Discharging mode of SC (Boost operation, $k = 1$)

## 5. Conclusions

The problem of developing a suitable controller for a buck–boost converter, which generates the charging and discharging current of the SC used in an electric vehicle, was studied, in order to satisfy the following requirements: (i) monitoring of the supercapacitor current up to its reference and (ii) asymptotic stability of the closed-loop system. The system studied consisted of a supercapacitor connected to the DC bus through a buck–boost power converter reversible in the current. The controller was obtained from the nonlinear averaged model (6) using an integral sliding mode control. Using both formal analysis and simulation and experimental results, it was shown that the obtained controller achieved the performances for which it was designed.

For future work, the focus will be on the energy management system, whose objective will be to develop the SC current reference, taking into account the constraints of the load and of the main source, which is the fuel cell.

**Author Contributions:** This paper was developed by a research team from LGS Laboratory; conceptualization, methodology, and formal analysis, H.E.F. and Z.E.I.; software and validation, Z.E.I., A.L., and M.O.; investigation and resources, Z.E.I., F.Z.B., and K.G.; data curation, Z.E.I.; writing, F.Z.B. and K.G.; review and supervision, H.E.F.; All authors have read and agreed to the published version of the manuscript.

**Funding:** This research received no external funding.

**Acknowledgments:** We gratefully acknowledge the support of the Moroccan Ministry of Higher Education (MESRSFC) and the CNRST under grant number PPR/2015/36.

**Conflicts of Interest:** The authors have no conflicts of interest to declare.

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
