# Peer review of "Theoretical Design and Experimental Validation of a Nonlinear Controller for Energy Storage System Used in HEV"

_wevj, doi:10.3390/wevj11030049_

Round 1

Reviewer 1 Report

The manuscript describes the theoretical design and experimental validation of a nonlinear controller for energy storage system used in HEV, however, the following needs to be addressed clearly:

Abstract: 

  1. "This paper deals with the problem of nonlinear control of..."What exactly is problem? Authors needs to be specific on the problem statement.
  2. "It is clearly shown, using formal analysis, simulation and experimental results that the designed controller meets the objective". What indications defines "Meeting the objective"? Authors are vague and not specific, detailed, clear and quantified results are necessary. Please consider.

Storage System presentation and modeling:

Fig.3 The block diagram needs to be described properly by including summation block signs (i.e.+/-) 

Storage Sliding Mode Control and Stability

  1. Line 149, "where ... and ... are the sliding surface coefficient and ... is the current reference", What symbol represent the current reference?
  2. Line 165, the authors refer to "time derivative", do they mean that the system instantaneous or what? Please clarify.

Simulation & Experimental Results

  1. Line 191, How did the authors obtain the stated parameters on Table 1? What was the criterion for selecting such parameters? This needs to be clearly described in detail.
  2.  Line 193-7, why was two different switching frequencies were used and type of switches for both simulation and experimental study. Why not similar type of switches for simulation and experiment, as IGBT is not similar to a MOSFET? Please elaborate.

Conclusion

Please consider other choice of words: the word "perfect" means 100% tracking, based on the results the tracking is not exactly 100% unless you would consider fitting algorithm.

References

Authors use mixed reference styles, it is important to be consistent in choosing the reference style. Please fix and pay attention to the following references in terms of page numbers, style:

[4], [6], [7], [10], [11], [13], [14], [21], [24], [25], [26], [28]-[30].

Author Response

Manuscript ID: wevj-834789

Journal: WEVJ (ISSN 2032-6653)

Title: “Theoretical Design and Experimental Validation of a Nonlinear Controller for Energy Storage System used in HEV”

Author’s reply to reviewers

The authors are grateful to the Reviewers for their helpful comments. The paper has been revised in accordance with Reviewer’s comments and suggestions. The changes made are described in the following responses to each Reviewer comment. The changes are highlighted in the paper using red colour.

Reviewer 1.

Reviewer comment 1

  • "This paper deals with the problem of nonlinear control of..."What exactly is problem? Authors needs to be specific on the problem statement.
  • "It is clearly shown, using formal analysis, simulation and experimental results that the designed controller meets the objective". What indications defines "Meeting the objective"? Authors are vague and not specific, detailed, clear and quantified results are necessary. Please consider.

Author’s reply

To protect the HEV supercapacitors and extend its lifespan, it is necessary to control the charge and discharge current. This work deals with this problem by setting the objectives to know:

  1. i) Monitoring of the supercapacitor current up to its reference.
  2. ii) Asymptotic stability of the closed loop system.

The two objectives are achieved using an integral sliding mode control.

We added objective with red in the abstract.

Reviewer comment 2

Fig.3 The block diagram needs to be described properly by including summation block signs (i.e.+/-)

Author’s reply

We added the signs in the block diagram.

Reviewer comment 3

  • Line 149, "where ... and ... are the sliding surface coefficient and ... is the current reference", What symbol represent the current reference?
  • Line 165, the authors refer to "time derivative", do they mean that the system instantaneous or what? Please clarify.

Author’s reply

-           The symbol represent the current reference is

-           It is the derivative of the equation (11) to study the stability of the system.

We have taken into consideration the remarks 1 and 2.

Reviewer comment 4

  • Line 191, How did the authors obtain the stated parameters on Table 1? What was the criterion for selecting such parameters? This needs to be clearly described in detail.
  • Line 193-7, why was two different switching frequencies were used and type of switches for both simulation and experimental study. Why not similar type of switches for simulation and experiment, as IGBT is not similar to a MOSFET? Please elaborate.

Author’s reply

  • It is too long to explain in the body of the paper all considerations that prevailed in the selection of various components. Therefore, we will develop the calculations here to justify the parameter values:

The Buck boost converter works in Boost when the supercapacitors supply electrical energy to the DC bus and in Buck when the electrical energy is supplied to the SC in order to charge them.

In fact the inductance L is defined by the following expression:

For f=15 kHz and  and, we have. But we don't have this value in the laboratory; we took what is available to do the experimental validation. So, we took, and we measured the resistance RL of the inductance equal to.

For the SC parameters (Csc and Rsc) are given by the manufacturer Maxwell.

  • We did the simulation with MOSFET transistors but the practical bench which contains MOSFET transistors to carry out experimental tests is currently not available in the laboratory.

If you want, we can repeat the simulation with IGBT transistors for the system to be identical to the practice bench.

However, we did our best to make the simulation conditions as close as possible to the actual experimental conditions.

It's just a question of the availability of the material.

Now, it is worth noting that the problem studied in this paper is partly of applied nature, the problem also involves theoretical issues pertaining to nonlinear control of the system

The difference between  switching frequency due to the difference in the type of the transistors.

Reviewer comment 5

Please consider other choice of words: the word "perfect" means 100% tracking, based on the results the tracking is not exactly 100% unless you would consider fitting algorithm.

Author’s reply

We changed the perfect word by monitoring.

Reviewer comment 6

Authors use mixed reference styles, it is important to be consistent in choosing the reference style. Please fix and pay attention to the following references in terms of page numbers, style:

[4], [6], [7], [10], [11], [13], [14], [21], [24], [25], [26], [28]-[30].

Author’s reply

We made the same style references.

The author’s thank the reviewer's critic.

Reviewer 2 Report

The buck-boost control of SC is not very novel, and the control strategy is also widely studied. In order to better highlight the specific contribution of this article, the following questions need to be supplemented.

1、Add a comparison with the results of Reference 21 in the simulation.

2、Add control and measured results of increased demand current to both simulation and test results.

3、The enlarged part in the figure needs special explanation, and pay attention to the time axis.

Author Response

Manuscript ID: wevj-834789

Journal: WEVJ (ISSN 2032-6653)

Title: “Theoretical Design and Experimental Validation of a Nonlinear Controller for Energy Storage System used in HEV”

Author’s reply to reviewers

The authors are grateful to the Reviewers for their helpful comments. The paper has been revised in accordance with Reviewer’s comments and suggestions. The changes made are described in the following responses to each Reviewer comment. The changes are highlighted in the paper using red colour.

Reviewer 2.

Reviewer comment 1

Add a comparison with the results of Reference 21 in the simulation.

Author’s reply

We compared our results with the reference results [21], the changes are mentioned in the paper with red color on lines 275 to 279.

Reviewer comment 2

Add control and measured results of increased demand current to both simulation and test results.

Author’s reply

Our first objective is to control the isc current of charge and discharge of the superconductor, in order to recover braking energy or to assist at the peak of energy demand. Figures 6 and 10 of the simulation and figures 14 and 18 of the experimental show the two cases; charging and discharging the SC by a constant current which is well controlled by a robust control law.

As future works, the focus will be made on energy management system whose objective will be to develop the SC current reference taking into account the constraints of the load and of the main source which is the fuel cell.

Reviewer comment 3

The enlarged part in the figure needs special explanation, and pay attention to the time axis.

Author’s reply

It is too long to explain in the body of the paper all considerations that prevailed in the selection of various figures. For more details, the enlarged parts have been added to the figures to show the invisible parts of the results. So that researchers can draw inspiration from our work.

The explanations of some enlarged parts of his figures are added in red.

The author’s thank the reviewer's critic.
